# Biomarkers for Inner Ear Disorders: Scoping Review on the Role of Biomarkers in Hearing and Balance Disorders

**DOI:** 10.3390/diagnostics11010042

**Published:** 2020-12-29

**Authors:** Nahla A. Gomaa, Zaharadeen Jimoh, Sandra Campbell, Julianna K. Zenke, Agnieszka J. Szczepek

**Affiliations:** 1Division of Otolaryngology-Head & Neck Surgery, Faculty of Medicine & Dentistry, University of Alberta, Edmonton, AB T6G 2B7, Canada; zenke@ualberta.ca; 2Faculty of Medicine & Dentistry, University of Alberta, Edmonton, AB T6G 2R3, Canada; 3Faculty of Science, University of Alberta, Edmonton, AB T6G 2E9, Canada; jimoh@ualberta.ca; 4John W. Scott Health Sciences Library, University of Alberta, Edmonton, AB T6G 2R3, Canada; sandra.campbell@ualberta.ca; 5Department of Otorhinolaryngology, Head and Neck Surgery, Charité–Universitätsmedizin Berlin, Corporate Member of Freie Universität Berlin, Humboldt-Universität zu Berlin, and Berlin Institute of Health, 10117 Berlin, Germany; 6Faculty of Medicine and Health Sciences, University of Zielona Góra, 65-046 Zielona Góra, Poland

**Keywords:** biomarkers, inner ear, cochlea, audio-vestibular organ, peripheral blood biomarkers, functional biomarkers

## Abstract

The diagnostics of inner ear diseases are primarily functional, but there is a growing interest in inner ear biomarkers. The present scoping review aimed to elucidate gaps in the literature regarding the definition, classification system, and an overview of the potential uses of inner ear biomarkers. Relevant biomarkers were categorized, and their possible benefits were evaluated. The databases OVID Medline, EMBASE, EBSCO COINAHL, CA PLUS, WOS BIOSIS, WOS Core Collection, Proquest Dissertations, Theses Global, PROSPERO, Cochrane Library, and BASE were searched using the keywords “biomarker” and “inner ear”. Of the initially identified 1502 studies, 34 met the inclusion criteria. The identified biomarkers were classified into diagnostic, prognostic, therapeutic, and pathognomonic; many were detected only in the inner ear or temporal bone. The inner-ear-specific biomarkers detected in peripheral blood included otolin-1, prestin, and matrilin-1. Various serum antibodies correlated with inner ear diseases (e.g., anti-type II collagen, antinuclear antibodies, antibodies against cytomegalovirus). Further studies are advised to elucidate the clinical significance and diagnostic or prognostic usage of peripheral biomarkers for inner ear disorders, filling in the literature gaps with biomarkers pertinent to the otology clinical practice and integrating functional and molecular biomarkers. These may be the building blocks toward a well-structured guideline for diagnosing and managing some audio-vestibular disorders.

## 1. Introduction

The first reports of biological markers, or biomarkers, date back to the 1980s, with the discovery of a soluble form of a membrane-bound enzyme active in the biosynthesis of the carbohydrate moiety of glycoproteins and glycolipids, detectable in patients with breast cancer [1]. Later, McCarthy and Shugart defined biomarkers as “measurements of molecular, biochemical, or cellular level in either wild populations from contaminated habitats or organisms when exposed to pollutants, toxic chemicals” [2]. Biomarkers are thought to increase in response to the magnitude of exposure of the organisms to a contaminant. Reports from the National Research Council published as early as 1987 classified biomarkers according to the exposure, effect, and susceptibility [3]. The clinical utility of a biomarker is predicated on it being specific to an organ or disease process and easily measurable (e.g., serum), changing in response to disease, and distinguishing between healthy and diseased states. Depledge and Fossi reported a more comprehensive definition. They described biomarkers as a biochemical, cellular, physiological, or behavioral change that can be measured in tissues or at the level of the whole organism that reveals the exposure at/or the effects of one or more chemical pollutants [4].

In the eighties and nineties of the 20th-century, biomarker discovery dominated the fields of oncology, infectiology, cardiology, and immunology [5,6]. In 1979, McCabe focused on idiopathic bilateral sensorineural hearing loss progressing to profound loss over weeks or months, occasionally associated with hearing fluctuations. Based on the positive response to cortisone and cyclophosphamide treatment, McCabe defined this illness as “autoimmune sensorineural hearing loss” [7]. This finding was a breakthrough in otology, as the inner ear was thought to be immunologically privileged, with immunoglobulin concentrations of 1/1000 of the concentration of immunoglobulin in the cerebrospinal fluid.

Other reports followed, identifying otolin-1, a scaffolding protein expressed during otoconia, in cells of the vestibule, and cochlear formation. Furthermore, higher levels of otolin-1 were found in individuals with benign positional vertigo [8]. Exosomes, extracellular vesicles involved in intracellular mechanisms related to interstitial body fluids, are another example of biomarkers of inner ear disease [9]. Further examples of biomarkers related to inner ear pathology include serum-based markers. Testing for connexin 26/30 and pendrin in evaluating hearing loss [10], as well as measurements of heat shock proteins in the diagnosis of autoimmune diseases [11], illustrates the utility of biomarkers, pending differential specificity and sensitivity. 

Some of the analyzed trials were based upon lymphocyte migration inhibition assays and lymphocyte transformation tests initially included in the diagnostic approaches of inner ear diseases; however, their diagnostic accuracy is not well established [12]. That is, even though IL-1-alpha, IL-2, and TNF-α are essential in the initiation, modulation, and amplification of the immune response occurring in the endolymphatic sac [13,14]. Therefore, the diagnosis of autoimmune inner ear diseases is still based on the clinical history without robust diagnostic tests, and with 15 to 30% of patients having associated systemic autoimmune disorders [12]. Similarly, idiopathic sensorineural hearing loss has been associated with viral infections and demonstrated increased serum concentrations of antibodies against cytomegalovirus (CMV), herpes zoster, herpes simplex type I, influenza B, and mumps [15,16].

A healthy inner ear is essential for hearing and balance, maintaining a high quality of life, and, importantly, preventing dementia [17]. Despite the growing interest in inner ear biomarkers, there remains a gap in the literature concerning a concrete definition of inner ear biomarkers or a published classification system. This scoping review aims to clarify the origin and development of the literature examining otological biomarkers indicating inner ear diseases. It comprises a search of the literature and attempts to provide a framework for the classification of known inner ear markers, critically analyze the available data, and examine the evidence associated with the use of biomarkers for diagnostic and treatment purposes. Consequently, this review can inform further research within the field and provide a snapshot of the state of our current understanding of biomarkers of inner ear diseases. 

## 2. Materials and Methods

### 2.1. Search Strategy and Databases

A search was conducted by an expert searcher/librarian (SC) on the following databases: OVID Medline, Ovid EMBASE, EBSCO COINAHL, CA PLUS (Scifinder), WOS BIOSIS, WOS Core Collection, Proquest Dissertations, and Theses Global, PROSPERO, Cochrane Library, and BASE (Bielefeld Academic Search Engine), using a controlled vocabulary (e.g., MeSH or Emtree). The general keywords used were: Inner ear Protein markers, Oxidative damage, Immune response in the inner ear, Inner ear markers in the circulation, Hearing loss, and Balance disorders, coupled with specific keyword searches such as inner ear proteins, Anticytoplasmic Antibodies (ANCA), and T-lymphocytes. Searchers were adjusted for each database, and animal studies were excluded. The references were exported to a citation management system. All searches are reported in the Appendix A.

### 2.2. Study Eligibility, Inclusion, and Exclusion Criteria

The criteria for inclusion of studies in this review were: peer-reviewed studies; international publications; studies examining biomarkers relevant to the inner ear, hearing, or balance disorders; human studies; primary studies and literature reviews; diagnostic and interventional studies, and neurological and immunologic markers relevant to the inner ear; clinical studies, systematic reviews, meta-analyses, and Cochrane reviews. The exclusion criteria were: animal studies, case studies, opinion papers, non-peer-reviewed studies, neurobiological and immunological markers relevant to health problems outside the inner ear diseases, and genetic biomarkers. 

### 2.3. Screening Process

After duplicate removal, the title and abstract screening and full-text screening were completed by two independent reviewers. 

### 2.4. Data Extraction and Analysis

The extracted data included: author, year of publication, study design, level of evidence according to “Oxford Center for Evidence-Based Medicine” [18], biomarkers along with their source of detection, application of biomarkers (i.e., diagnosis, treatment). 

## 3. Results

One thousand five hundred and two studies were identified through the database search (Figure 1). Following duplicate removal, 1160 studies were included in the title and abstract screening. Of these, 987 studies were excluded, leaving 173 studies meeting the criteria for full-text screening. Through full-text screening, other 141 studies were excluded for various reasons, leaving 32 studies that were included in the review. Two additional studies were identified through reference tracking, bringing the total of reviewed publications to 34. 

Two reviewers screened the studies with inter-rater reliability; the Kappa score was 70.5 between the reviewers. 

### 3.1. Included Studies

Studies published between 1999 and 2020 were thoroughly assessed for conclusions. They were also critically appraised for the level of evidence, any weakness, biases, and whether the study objectives were met. Of the 34 articles that were included, 7 were review articles, 1 was a research guidance paper, 5 were retrospective studies, 9 were prospective observations, and 12 were prospective correlational or randomized clinical trials.

On the basis of utility, 23 studies [33.7%] looked at molecular markers, 7 of which were review articles; 10 studies examined functional markers, 7 of which were review articles; and 1 study investigated both types of markers. We determined the significant variability in the level of evidence, ranging from 1a to 5. Furthermore, the purpose of the studies varied and included diagnostic, pathophysiologic, therapeutic, or predictive aims, as shown in Table 1, Table 2, and Table 3. 

The identified biomarkers were also classified on the basis of their clinical application, namely diagnostic, prognostic, or pathologic. Fifty-five molecular (Table 1) and 10 functional biomarkers (Table 2) related to inner ear pathology were identified in the original articles, added to many other briefly mentioned biomarkers included in review articles (Table 3). 

### 3.2. Biomarker Classification

Of the functional biomarkers, seven were diagnostic, three predictive, and two were prognostic. Of the molecular studies, the majority were identified as having diagnostic applications; a single prognostic study was found, and a single pathological biomarker. The molecular biomarkers in review studies showed more pathophysiologic, pathogenic, and diagnostic values. We noticed that although some studies had a higher level of evidence, based on the “Oxford Centre for Evidence-Based Medicine classification” [18], this classification acknowledged the proposed methodology, leaving these studies with unmet pre-set objectives. Therefore, we added another classification based on a verdict from the reviewers on whether the study met its own pre-set objectives or not. 

#### 3.2.1. Molecular Biomarkers [Table 1 and 3]

##### Inner Ear-Specific Protein Biomarkers that Can Be Detected in Peripheral Blood, Plasma, or Serum

##### Protein Biomarkers

**Mulry and Parham** included a very comprehensive description of the roles of protein biomarkers [19]. The proteomic analysis enabled the identification of many specific inner ear proteins, some of which are detectable in peripheral blood and others, found explicitly within the inner ear. The proteins that can make their way through the blood-labyrinthine barrier and be detected in the blood are otolin-1, prestin, and matriline.

**Otolin-1** is a 70 k Da glycoprotein, linked to inner ear sensory hair cells and the tectorial membrane. It assists in scaffolding calcium carbonate deposition into otoconia with otoconin-90. Hence its presence in many areas of the inner ear, including the supporting cells of the organ of Corti, striae vascularis, and semicircular canals [20]. Significantly increased levels have been found in inner ear trauma cases in mastoidectomies with a positive correlation with drilling time, confirming its role [21]. Additionally, Sacks and Parham found a strong correlation between otolin-1 levels, benign paroxysmal positional vertigo (BPPV), and osteoporosis [22]. Despite a close relationship between the two proteins, there is no evidence that Otoconin-90 could be measured in peripheral blood, while otolin-1 has been precisely measured in serum using an enzyme-linked immunosorbent assay (ELISA) [8,20]. 

**Prestin** is an 80 kDa motor protein of the cochlear outer hair cells (OHC). The presence of prestin in serum could serve as an indicator of OHC damage. Parham first suggested this while studying the hearing loss caused by noise exposure in Wistar rats [23]. This observation was later confirmed in a human study of idiopathic sudden sensorineural hearing loss (ISSHL) by Sun et al. [24]. This latter study also suggested the prognostic role of prestin level reduction in cases of ISSHL with effective treatment and restoration of hearing thresholds. 

**Matrilin-1** is a 148 kDa cartilage-specific protein, found mainly in the upper airway cartilage (nasal and tracheal) [25], proved to be measured in serum in correlation with cartilage inflammation, and could be detected in the serum of patients with relapsing chondritis (RC) [26].

##### Biomarkers of Inflammation

##### Antibody Biomarkers

**Anti-type II collagen antibodies.** Arnaud et al. emphasized that antibodies against type II collagen were detected in 33% of sera of patients with relapsing polychondritis [26]. 

**Antinuclear antibodies (ANA)-**ANA serum titers are 21–28% higher in patients with Meniere’s disease (MD) [27]. In addition, there is an increasing incidence of connective tissue diseases (CTD), including rheumatoid arthritis, ankylosing spondylitis, and systemic lupus erythematosus in patients with MD. Moreover, patients with mixed connective tissue disease (MCTD) who suffered from sudden sensorineural hearing loss (SNHL) had higher serum levels of anti-U1RNP, anti-endothelial cell antibodies, and IgG type anticardiolipin antibodies than MCTD patients without SNHL. The serum concentration of interferon-γ and tumor necrosis factor-α was higher in MCTD patients with SNHL than patients without SNHL, while the absolute number of natural regulatory T cells (CD4 + CD25) was lower than in patients without SNHL [28]. On the other hand, a bilateral progressive sensorineural hearing loss, which can be autoimmune inner ear disease (AIED), can coexist with another autoimmune disease in 15–30% of patients [29], which could, in turn, be associated with other antibodies, such as anti-centromere antibodies [30]. 

There is a consensus of evidence on the correlation between mixed connective tissue diseases (MCTD) and hearing loss. Hajas et al. estimated that 46.4% of MCTD patients had SNHL, with no correlation to the duration or age of onset of MCTD [28]. While 30.3% of patients had audiometrically documented SNHL, 69.7% had audiometrically-detectable SNHL without symptoms. Audiometric patterns in this study showed pantonal hearing loss in 33.3% and high-frequency loss in 66. 6% of patients with SNHL, present in 9.0%, and vertigo in 6.0% of MCTD patients. In MCTD, there is evidence of antigenic structural modifications reflected by T-cell-dependent B-cell responses driven by antigenic stimulation against self-antigens [31]. Antinuclear antibodies and anti-U1RNP were found to be characteristic of the disease, and other antibodies such as anticardiolipin (aCL) or anti-endothelial cell antibodies (AECA) may determine the clinical symptoms and the disease course [32,33]. 

Alawieh et al. reviewed the proteomic profile of the inner ear, resulting in the identification of about 50 proteins. The profiling of plasma samples obtained from MD patients showed overexpression of factor H and B, fibrinogen α and γ, β actin, and pigment epithelium-derived factor proteins [34]. Concurrently, there was an underexpression of β-2 lipoprotein I, vitamin D binding proteins, and apo-lipoprotein I compared with controls. Due to the low sample size, further studies are needed for confirmation [35]. 

Lee et al. studied inflammatory biomarkers and autoimmune responses in bilateral sudden sensorineural hearing loss (S-SNHL) [30]. They found that while erythrocyte sedimentation rate (ESR) and C-reactive protein (CRP) represented disease activity and response to treatment, a high count of white blood cells (WBC) was rare in their cohorts. In a second step, they found that animal inner ear antigens have reacted with sera from patients with bilateral S-SNHL. Their results were in agreement with Yukawa et al. [36] and Harris et al. [37], both of which specified inner ear protein molecular weights to be in the 42, 58, 68, and 72 kDa ranges. Haasse and Prassad added cytokines (TNF-α & interleukin-6) and enzymes (synthase) to the inflammation biomarkers and confirmed the importance of C-reactive protein (CRP) [38]. They also emphasized that IL-6 is a critical proinflammatory cytokine. Contrary to these findings, Süslü et al. concluded that, due to a low titer, antibodies against HSP 70, TNF-α ESR, and ANA did not offer clinically useful information for the treatment of SSNH [39]. Although anti-phospholipid antibodies were initially assumed to play a role in activating microthrombi formation in the inner ear in cases of SSNHL [40], none of the 60 patients included in the study of Süslü et al. had anti-phospholipid IgM or IgG antibodies.

Even though the work of Betancur et al. aimed to have a prospective correlational design and included a large number of subjects (113, 491), a retrospective design was used, with limited or no access to all clinical data, and it was not clear how geographical and racial differences in the prevalence of rare ANA patterns were balanced [41]. A possibility of misclassification exists, as a specialist was not always involved in the diagnostic process. Moreover, the study did not confirm anti-NuMA specificity (NuMA-1 or Ant-HsEg5) by immune blotting of sera. 

In an observational study in 11 children without a control group, Berti et al. conducted an analysis based on limited alternatives of organ non-specific and hypothetically organ-specific autoantibodies, including antibodies against Cogan-peptide, connexin26, DEP1/CD148, and reovirus [42]. Only two children who tested positive for the presence of auto-antibodies developed juvenile hearing loss before the age of 16. 

##### Antiviral Antibodies

The antibodies found in the serum of patients with idiopathic sudden deafness included antibodies against CMV, herpes zoster, herpes simplex type 1, influenza B, and mumps [15,16]. Nonetheless, there was no correlation between the antibody titer and hearing loss [16], and the Henle–Koch postulates could not be satisfied [15], making the direct association between the viral infection and the hearing loss uncertain. 

##### Other Biomarkers

**Complement Factor H** is a glycoprotein that plays a role in the alternative complement activation pathway. However, a mutation in a gene encoding complement factor H was linked to membranoproliferative glomerulonephritis [43], hemolytic uremic syndrome [44], and age-related macular degeneration [45]. It was also elevated in otitis media with effusion [46], which was recently correlated with MD. 

**Beta-2 glycoprotein 1 (beta-2GP1)** is another glycoprotein that binds to negatively charged substances (heparin, dextran sulfate), preventing the activation of the intrinsic blood coagulation cascade by interacting with phospholipids on the surface of damaged cells. Auto-antibodies targeting beta-2GP1 were recently detected in patients with sensorineural hearing loss [47].

**Beta-actin** is a structural protein essential for the cytoskeleton’s integrity and indispensable for the structure of stereocilia in the hair cells. Mutations that may alter actin filaments’ polymerization in the cytoskeleton and other organelles have been correlated to autosomal dominant hearing loss and dystonia with non-syndromic hearing loss [48,49].

**Vasopressin** is a hormone that regulates the osmotic pressure of bodily fluids. A retrospective study showed higher vasopressin levels in the acute stage of Meniere’s Disease [50]. This is in agreement with Kumagami et al., who reported that vasopressin might decrease fluid reabsorption in the endolymphatic sac, inducing endolymphatic hydrops in guinea pigs, suggesting that the release of vasopressin can provoke a vertigo attack in Meniere’s disease [51]. Another animal study demonstrated that plasma vasopressin levels decrease after compression of the inner ear pressure [52].

**Vitamin D binding protein (VDBP)** is also known as Gc-globulin (group-specific component globulin) known as a scavenger of actin, thus protecting cells from the toxic effect of intravascular actin polymerization [53]. Mechanistically, VDBP has been shown to reduce platelet aggregation and to prolong the coagulation time ex vivo [54].

**Glucocorticoids and brain-derived neurotrophic factor (BDNF)** have been advocated by Rüttiger et al. [55]. Human serum contains BDNF at much higher concentrations (factor 200) than human plasma [56,57], and it was more linked to hyperacusis and tinnitus. 

##### Inner-Ear-Specific Biomarkers Detected in the Perilymph or Inner Ear Structures

**Cochlin** is a protein expressed in the cochlea and vestibule [58], with a remarkable role in diagnosing the perilymphatic fistula, with a sensitivity and specificity of 86.4% and 100%, respectively [59]. Moreover, cochlin was found to be overexpressed in MD [60] and autoimmune inner ear disease [61]. Despite evidence of its role, cochlin was always detected in the perilymph, utricle, or crista ampullaris. Other proteins that may be found in the inner ear are prestin, otoancorin, otogelin, α-tectorin, β-tectorin, and otoconin-90, in addition to otospiralin [62], otoraplin [19,63], and oncomodulin [64,65,66]

**Heat Shock Proteins (HSP)** is a family of cytoplasmic chaperon proteins responsible for other proteins’ folding process. Pennisi et al. identified HSP 90 as an essential chaperone protein involved in stress reactions [67]. Rather than being involved in de novo protein folding, HSP 90 is required for protein maturation. Schmitt et al. conducted an observational study for heat shock proteins in 39 perilymph samples collected during ear surgery, highlighting the difficulty in obtaining perilymph samples from the control subjects [11]. The same challenge was noted in temporal bone studies of the sensorineural effects of chronic otitis media through staining for 27E10, a monocyte-macrophage biomarker. Despite challenges related to sample accessibility, Jókay et al. suggested the scala tympani as the site of lymphocyte-macrophage interaction, causing SNHL [68]. 

##### Biomarkers of Oxidative Stress [Damage]

Antibodies against cardiolipin and Saccharomyces cerevisiae (ASCA) were studied by Haase and Prassad, who classified oxidative stress biomarkers into lipid peroxidation (including malondialdehyde and F2-isoprostane), DNA adduct (8-hydroxydeoxyguanosine), and protein modification (3-nitrotyrosine protein carbonyls) [38]. Other studies showed differences between oxidative stress biomarkers in cases of noise-induced hearing loss (NIHL) and other cochlear damage mechanisms. NIHL reflects free radical activity; therefore, its stress biomarkers include malondialdehyde, 4-hydroxynonenal, nitrotyrosine, and inducible NO synthase (iNOS) [69]. Other proteins such as cytochrome C and enzymes such as caspases may also be involved as mediators of apoptosis and contribute to cochlear nerve fiber degeneration. 

In comparison, idiopathic sensorineural hearing loss is characterized by a “two-hit” mechanistic response, namely, cochlear ischemia, followed by reperfusion injury. Excess excitotoxicity caused by glutamate appears to initiate the primary damage by hydroxyl radicals, and nitric oxide aggravates cochlear perfusion and swelling. The biomarkers related to tinnitus are peroxynitrite, nuclear factor kappa beta (NF-kB), and elevated intracellular calcium [70], with increased markers of proinflammatory activity such as cytokines, tumor necrosis factor-alpha (TNF-α), and interleukin-1 (IL-1) [71].

Drug-induced ototoxicity involves increased cochlear levels of several biomarkers, including malondialdehyde, which indicates the activity of free radicals. It is also an intracellular aldehyde formed by the decomposition of lipid peroxides (unstable derivatives from the metabolism of polyunsaturated fatty acids), xanthine oxidase, and NO elevation. Levels of endogenous antioxidants such as glutathione and antioxidant enzymes such as superoxide dismutase, catalase, and glutathione peroxidase are decreased, whereas a major regulator of antioxidant enzyme expression, the transcription factor nuclear factor-erythroid 2-related factor-2 (Nrf2), protectively upregulates these enzymes against ototoxicity [72]. Concerning toxicity-related vestibular dysfunction, the findings follow the same path. The expression of proinflammatory cytokines and NF-kB is elevated in animal models of vestibular hair cell damage [73]. Genetics is also beginning to play an increasing role in understanding the response to oxidative damage in the inner ear. 

Evans and Halliwell contributed to our understanding of the pathophysiology of oxidative stress and the role of antioxidants in the prevention and treatment of inner ear diseases [74], but so have others. Okano explained the inner ear’s innate immunity, including inner ear macrophages and immune-mediated inner ear disease [75], and Hasse and Prasad further described research guidelines addressing the necessity for standardized techniques of samples and analytical technology, and highlighted cost as a concern [38]. 

##### Other Biomarkers

**B7 homolog 1 (B7-H1)**, also known as programmed cell death ligand 1 (PD-L1) or as a cluster of differentiation 274 (CD274), is a surface protein involved in the suppression of adaptive immunity. Archibald et al. studied B7-H1 on CD8+, CD3+, and CD4+ lymphocytes in vestibular schwannoma fresh frozen specimens during surgical removal [76]. Since B7-H1 is typically not expressed in healthy tissues, the study lays the foundation for the potential use of a monoclonal antibody to block B7-H1. A shortcoming of the study is the fact that only 33% of the tumors strongly expressed B7-H1, and 48% showed mild positive expression, in addition to the challenge of obtaining tumor tissue in vivo for immunohistochemistry. 

**Receptor tyrosine-protein kinases ErbB2 (proto-oncogen) and ErbB3 (a dimer partner of ErbB2)** were implicated in tumor growth, proliferation, and chemotherapeutic resistance. An observational study by Edvardsson Rasmussen et al. shows that the presence of ErbB2 and ErbB3 in the perilymph may be relevant to vestibular schwannoma tumor size [77]. A retrospective study using immunohistochemistry showed that vestibular schwannoma cells produce ErbB2 and B3, while the Scarpas ganglion produces neuregulin, that might interact with ErbB2 and ErbB3 [78]

**Inner ear melanocytes.** Barozzi et al. reviewed the role of inner ear melanocytes in audiovestibular diseases [79]. However, the authors could not establish a relationship between melanocytes and labyrinthine disorders and recommended further research. 

#### 3.2.2. Functional Biomarkers [Tables 2 and 3]

Notably, the number of papers featuring functional biomarkers in this search was very limited (Table 2). Rüttiger et al. suggest that biomarkers should be a metric or an indicator of a physiological disease state. Based on this definition, evoked potentials related to hearing and balance—including otoacoustic emissions (OAEs), SP/AP ration in MD, auditory brainstem responses (ABRs), middle latency responses, cortical auditory evoked potentials (CAEP), and vestibular evoked myogenic potentials for balance—should be considered biomarkers [55]. 

**Event-related potential P1, N1, P2 Complex (CAEP).** Campbell et al. focused on applying the event-related potential for auditory neuropathy spectrum disorders [80], while Mostafa et al. were trying to use it as an indicator of better amplification, comparing between hearing aids and cochlear implants [81]. 

**Brainstem auditory evoked responses (ABR).** Counter and Buchanan confirmed evidence for using the primary three waves I, III, V of the ABR as a biomarker for sensorineural hearing loss caused by lead poisoning, with a good correlation between ABR and audiometric thresholds, in addition to delayed absolute latencies of the main three waves without effect on inter-wave intervals, a finding that is in agreement with sensorineural hearing loss effects [82]. The study, however, did not include a with a control group. Additionally, there was no correlation study between ABR or any neural correlates and the severity of lead poisoning or blood lead levels. 

Dewey found merit in adding fMRI to the ABR while testing noise exposure effects [83]. On the ABR side, they used I/V amplitude ratio as a biomarker across low and high noise exposure groups. fMRI, however, added the benefit of the localization of the level of damage [84].

**Functional magnetic resonance imaging (fMRI).** Deshpande et al. found a correlation between the activation of the angular gyrus, supramarginal gyrus, middle temporal gyrus, precuneus, medial frontal gyrus, orbital gyrus, cingulate gyrus, subgyral, and middle occipital gyrus with post-cochlear implant speech language-auditory performance in children with hearing loss implanted before 36 months [83]. However, using anesthetics (Propofol) in children is known to contribute to the variability of imaging results, limiting the interpretability of these results. Additionally, there is rapid development in the anatomical microstructure of the brains of young children up to 36 months old, which added to the complexity of performing fMRI studies in this age group [85]. 

**A temporary threshold shift (TTS).** Bekesy audiometry was used in 1980 to test noise perception changes, as the noise level is changed in ascending and descending ways [86]. Feuerstein et al. could not prove an increase in the temporary threshold shift difference after acute noise exposure [87], whereas Moshammer et al. used the TTS in predicting the duration of noise exposure [88]. Considering that the measurement of OAEs allows for the functional evaluation of the OHCs, which are the primary targets of NIHL, added to the fact that OAE suppression by contralateral activation of the medial olivocochlear reflex was proposed a predictor of the sensitivity of workers to NIHL, OAE’s occupational screening received much more attention [89].

**Elelctrovestibulography (EVestG).** Although Garrett et al. [90] concluded that the separation of the response into acceleration and deceleration might be beneficial in distinguishing between Benign Paroxysmal Positional Vertigo (BPPV) and Meniere’s disease, the results suggested that the DC component warrants further analysis [91].

## 4. Discussion

This scoping review evaluated the clinical applications of the biomarkers related to hearing and balance disorders. Of particular interest were the biomarkers with the potential to be used by the otolaryngologist in a clinical setting. Parham et al. suggested that an ideal biomarker should be specific, detectable, measurable, and meaningful [8]. To date, serum-identified markers in idiopathic sudden sensorineural hearing loss (ISSHL) include erythrocyte sedimentation rate, interleukin 6, C-reactive protein, and various autoimmune and prothrombotic markers [39,97]. These markers are non-specific and yielded limited and inconsistent information regarding disease pathophysiology. 

More than a hundred markers were identified in the selected studies, including review articles. That excludes genetic biomarkers and additional markers detected in animal studies (yet to be considered in humans), with little impact in the field of otology and consequently deemed outside the scope of our work.

Through the review, it became evident that there are many natural ways to classify the biomarkers of inner ear dysfunction. Firstly, one may classify these markers according to the dichotomy molecular versus functional. The literature revealed that there are relatively few studies looking at functional biomarkers or relating the evoked potential and imaging to the category of “functional biomarkers”. This knowledge gap presents an area for further research. Another way to classify these biomarkers is through their clinical utility in terms of prognostic, diagnostic, and treatment value. However, given this novel area of research, further studies are needed to demonstrate these biomarkers’ practical utility in a clinical setting and their importance on the point of care. Moreover, many currently identified biomarkers are impractical in a clinical context, being non-specific (e.g., beta-actin or proinflammatory cytokines), invasive to collect, or requiring specialized equipment not readily available. Finally, the biomarkers of inner ear disease may be either inner-ear-specific or inner-ear-non-specific. Unfortunately, to date, inner-ear-specific markers are difficult to measure in patients.

The “Oxford Center for Evidence-Based Medicine” [18] is an excellent tool for sifting through the studies based on their methodology design. Nevertheless, we noticed that some studies, especially those on molecular biomarkers, had great ambitions in terms of the number of biomarkers at the beginning of the study, or a complicated method for accessing the samples. That has resulted in incomplete study outcomes or early end-points for these studies. Therefore, we added another classification based on whether these studies meet their own pre-set objectives: the authors of the papers declared that the objectives had not been met, or the objectives were partially met with recommendations to expand on the sample size or to complete additional biomarkers data. The two senior authors were responsible for the decision making on the 34 studies selected for full data analysis, 18 (53%) were considered to have met their objectives, while ten (29.4%) did not, and six (17.6%) did so only partially. The main reasons for not meeting or partially meeting the objectives were the broadness of the objectives, or the fact that the biomarkers were detected mainly in the inner ear, which makes them difficult to use. Moreover, it is hard to prove their specificity due to the invasiveness of the sampling procedure, requiring a highly dexterous surgical accessibility to the inner ear; and carrying a higher risk percentage of sample pollution or the mixing of endo- and perilymph in some research protocols.

Parham et al. [8] spear-headed the introduction of the concept of blood-based biomarkers for inner-ear disorders, initially for vertigo as proof of concept, and demonstrated that patients with benign paroxysmal positional vertigo had significantly higher levels of an inner-ear protein, otolin-1, than controls. Many authors expressed the rarity of reliable blood markers to reflect otologic orders, with the exceptions of genetic biomarkers such as connexin 26/30 in hearing loss cases [98,99] or the use of heat shock proteins for the same reason [100], given the limitation of its specificity. It was then concluded that no inner-ear-specific serum biomarkers had been approved for use in clinical otology. 

Our review shows aggregated data that prove the detection of inner-ear-specific biomarkers in peripheral blood when they cross the labyrinthine blood barrier [22,25]. However, these studies are not supported by enough evidence to prove these specific biomarkers’ relevance to the point of care in clinical practice. These studies are also dispersed within the vast number of biomarkers. Inner ear proteins, including otolin-1, prestin, and matrilin-1 have shown consistency in their presence in peripheral blood associated with auditory disorders [27,48,53,54]. Inflammatory markers such as CRP and ESR, as well as ANA, p-ANCA, c-ANCA, Anti-DNA screening, C3, and C4, including IgA, IgG, immunoglobulin heavy or light chains, are also measurable, yet their false-positive results in cases of systemic autoimmune disease have to be taken into consideration. Furthermore, any non-inner-ear-related or non-autoimmune inflammatory condition (e.g., a common cold), as well as the aging process, could influence the inflammatory markers’ diagnostic value for inner ear disorders. Various antibodies in serum correlated with inner ear diseases (e.g., anti-type II collagen, antinuclear antibodies, antibodies against cytomegalovirus, herpes zoster, herpes simplex type 1, influenza, and mumps). Further studies will elucidate the clinical significance and diagnostic or prognostic usage of biomarkers for inner ear disorders.

There is also a tendency toward Meniere’s disease or vertigo-specific biomarkers [29,32], namely beta-2 glycoprotein, beta-actin, complement factor-H, and vitamin D binding protein. Looking into the few studies that probed the integration of functional biomarkers into the diagnostics and prognostics of audio-vestibular disorders is clinically promising. These functional biomarkers ranged from evoked potentials to fMRI, which is currently research-based. Given their specificity to audio-vestibular functions, the feasibility of measuring, and non-invasiveness, they may constitute a perfect supporting tool to the relatively non-specific molecular biomarkers. 

### 4.1. Study Limitations

Myriads of research works have been done on disease markers associated with hearing and balance, inner ear, or audiovestibular disorders. However, an inherent limitation of the search-based studies is the fact that the search keywords limit them. In this particular domain, not all researchers mention the word “biomarkers” or “inner ear biological markers” in their publications’ keywords, study titles, or conclusions. 

### 4.2. Future Directions

Due to the importance of each group of biomarkers, the authors would like to expand into dedicated systematic reviews that address each group separately, such as the use and utility of genetic biomarkers or other molecular biomarkers in specific inner ear diseases and whether this has a role in improving the point of care for a specific group of patients or improving sequelae. 

Integrating functional biomarkers and including evoked potentials of hearing (e.g., auditory evoked potentials) and balance-related tests (caloric test results, vestibular evoked myogenic potentials [VEMPs], posturography, and rotational chair responses) may enrich the diagnostic/prognostic research evidence in specific audiovestibular entities, including Meniere’s disease.

## 5. Conclusions

There are plenty of scattered studies with an abundance of molecular indicators of disease or function. This work is a trial to arrange the available data into inner-ear-specific and non-specific biomarkers. We found that there is a stronger tendency to find the keyword “biomarker” in molecular research than in functional studies, making it difficult for the latter to be identified in a similar review. The majority of biomarkers measured in peripheral blood are inflammatory (e.g., cytokines) or not restricted to the inner ear (e.g., beta-actin) and, therefore, non-specific. The identification of inner-ear-specific small molecules with the potential to cross the labyrinthine blood barrier to peripheral blood should be a priority in the search for inner ear biomarkers (Figure 2). There is merit in using functional biomarkers to support molecular biomarkers in inner ear studies. Revisiting the classifications of biomarkers, filling the gaps in knowledge with studies dedicated to the most clinically-applicable ones, and comparing this to the combined usage of molecular-functional biomarkers may be the way to go in the future of this research. Additionally, revisiting the evidence-based classification systems of research to reflect a more outcome-based consensus and comparing the studies’ outcomes to their pre-set objectives is in demand. 

## Figures and Tables

**Figure 1 diagnostics-11-00042-f001:**
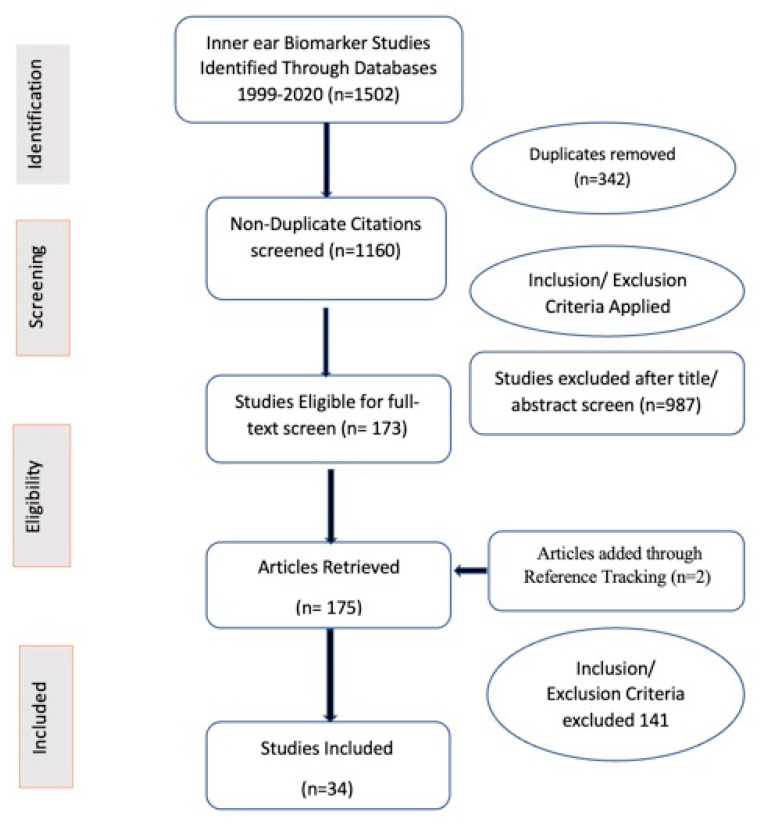
PRISMA (Preferred Reporting Items for Systematic Reviews and Meta-Analyses) diagram for article selection.

**Figure 2 diagnostics-11-00042-f002:**
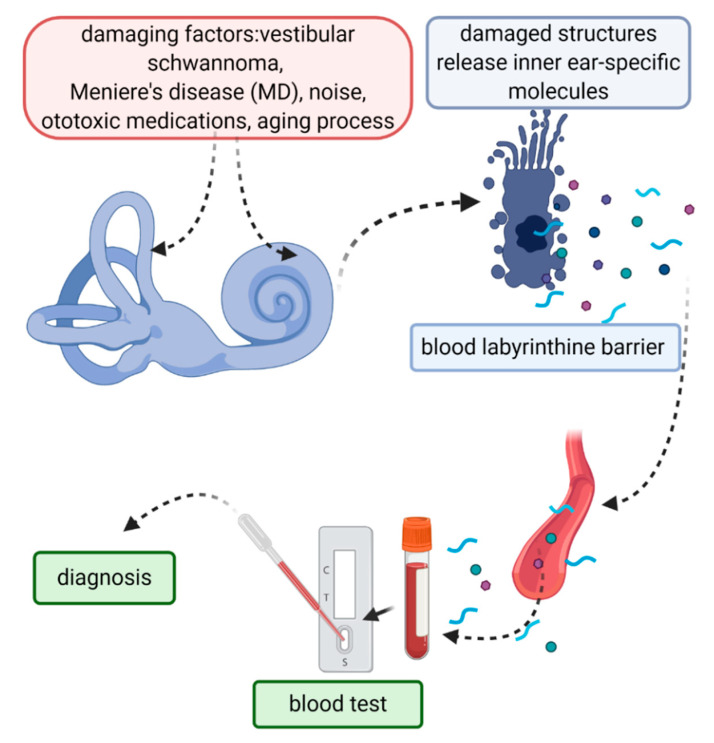
The identification of inner-ear-specific and disease-specific molecular biomarkers should in the future improve the otologic diagnosis and prognosis. Created with BioRender.com.

**Table 1 diagnostics-11-00042-t001:** Molecular biomarkers related to the inner ear in humans.

Author	Year	Study Type	Level of Evidence	Met Objectives	Biomarker	Key Source	Category	Ref.
Archibald et al.	2010	Blind retrospective chart review	3b	Yes	B7-H1 expression in lymphocytes	Fresh frozen vestibular schwannoma Tissue	Diagnostic	[76]
Edvardsson Rasmussen et al.	2018	Observational	4	Yes	Alpha-2-HS-Glycoprotein	Perilymph aspirated through round window membrane	Prognostic	[77]
Schmitt et al. *	2018	Observational	3b	Partially	Heat shock proteins	Perilymph & Cochl.T	Diagnostic	[11]
Lee et al. **	2017	Observational	1b	Yes	Autoimmunity; elevated sera CRP & ESR in addition to many other proteins specific for autoimmune inner ear disease, or sudden SNHL	Blood	Diagnostic	[30]
Kim et al. ***	2014	Prospective correlational	3b	partially	Neuro-immunology antigens involved in immune reactions	Endolymphatic sac luminal fluid from 3 patients & peripheral blood from 10 patients	Diagnostic	[92]
Betancur et al.	2018	Prospective correlational	1b	Yes	Anti-centrosome antibodies, MSA-2 CENTP-F/MSA-3, Nu-MA/MSA-1	Serum	Diagnostic	[41]
Hajas et al. ****	2009	Prospective correlational	2b	Partially	T cell count, IgG antibodies Anti-endothelial cell antibodies [AECA]; High levels of anti- U1RNP, IgG type aCL & AECA	Plasma & Serum	Diagnostic	[28]
Ikezono et al.	2018	Prospective correlational cohort	2b	Yes	Perilymph specific protein Cochlin-tomoprotein [CTP]	MEL [Middle ear lavage] & Peripheral blood [Plasma & Serum	Diagnostic	[59]
Berti et al. *****	2013	Observational study	3b	No	Inner ear autoantibodies	Serum	Prognostic	[42]
Sacks & Parham	2015	Prospective pilot clinical trial	2b	No	Otolin-1	Serum	Diagnostic	[22]
Haase & Prasad ******	2016	Research guidance papers	1a	Yes	Biomarkers of oxidative stress; Biomarkers of inflammation	Peripheral blood	Diagnostic	[38]
Aoki et al.	2007	Retrospective cohort	2b	No	Plasma osmolality; plasma Vasopressin levels	Plasma	Diagnostic	[50]
Parham et al.	2014	Retrospective controlled cohort	3b	No	Otolin-1	Serum	Diagnostic	[8]
Jókay et al.	2001	Retrospective observational	2b	Yes	Staining for 27E10 in chronic otitis media	Temporal bones from autopsies	Pathogenic	[68]
Hansen & Linthicum	2004	Retrospective observational	4	Yes	ERB2 & E3, Neuregulin	Pathological specimens of vestibular schwannoma	Diagnostic	[78]
Chiarella et al.	2012	Cohort	2b	Partially	Beta 2 glycoprotein, beta actin, complement factor-H, vit. D binding protein	Plasma	Diagnostic	[35]

* Heat shock proteins were detected in perilymph samples collected during inner ear surgery. They included HSP 4, 5, 6, 8 HS 27kDa, HS 90 α & β, Alpha-crystalline β-Chain and Endoplasmin. ** Identified biomarkers in bilateral S-SNHL, looking for neutrophil differentials, CRP and ESR [inflammatory biomarkers as well as ANA, p-ANCA, c-ANCA, Anti-DNA screening, C3, and C4. They also identified specific proteins in the sera of those patients, including IgA, IgG, Immunoglobulin heavy chains, chains A, G, H, and L. *** 9 proteins identified in the inner sac luminal fluid, including AF1, nonallergic heavy chain, IgM, IgG, Immunoglobulin, Kappa light chain, and interferon regulatory factor. **** Serum levels of Anti- U1RNP, anti-SSA, anti-SSB, anti-Jo1, anti-Scl70, anti-dsDNA, Cogent, anti-ß2-glycloprotein I (anti-ß2-GPI; IgG, IgM, and IgA) antibodies, and aCL=Anticadiolipin, Plasma levels of anti-endothelial cell autoantibodies (AECA). Serum soluble Cytokines, Serum interferon, Tumor necrosis-α [TNF-α], Interferon-γ [IF-γ] & Interleukin-10 [IL-10] raised the possibility that anti-TNF-α monoclonal antibodies or TNF-α receptor blocker may be useful in the therapy of SNHL. ***** Peripheral blood Inner ear antibodies included ANA; Anticardiolipin Antibody [aCL], Antineutrophil Cytoplasmic Antibody [ANCA]; anti-Saccharomyces cerevisiae antibodies (ASCA). ****** Oxidative damage biomarkers included lipid peroxidation [Malondialdhyde, F-2 isoprostane], DNA adduct [8-hydroxydeoxyguanosin], and Protein modification [3-Nitrotyrosin Protein Carbonyls]; Inflammation biomarkers included Cytokines [TNF-α & Interlerlukin-6]; Protein [CRP] and Enzymes [Synthase]; Excitatory neurotransmitters [Glutamate]; Lipid-soluble antioxidant [α-Tocopherol] and water-soluble antioxidants [Glutathione].

**Table 2 diagnostics-11-00042-t002:** Functional biomarkers related to the inner ear in humans.

Author	Year	Study Type	Level of Evidence	Met Objectives	BioMarker	Category	Ref
Mostafa et al.	2014	Correlational cohort	1b	Partially	P1 CAEP	Diagnostic	[81]
Dimitrijevic	2016	Correlational thesis	2b	No	Envelope following response	Diagnostic/Prognostic	[93]
Counter & Buchanan	2002	Correlational	2b	No	Brainstem auditory evoked responses	Diagnostic	[82]
Campbell	2011	Descriptive	4	No	P1 component of the cortical auditory evoked potential	Treatment	[80]
Dewey et al.	2018	Observational	1b	No	Regions of Interest (ROIs) on fMRI	Diagnostic	[84]
					Brainstem auditory evoked responses	Diagnostic	
Feuerstein et al.	2015	Observational	5	No	Temporal Threshold Shift (TTS)	Predictive	[88]
Moshammer et al.	2015	Pre- and post, correlational	1c	Yes	Temporal Threshold Shift (TTS)	Diagnostic	[88]
Coffey et al.	2016	Prospective observational	2b	Yes	Auditory Frequency Following Response (FFR)	Diagnostic	[94]
Choi et al.	2017	Secondary analysis of a double-blinded randomized clinical trial	2b	Yes	Electrode impedance fluctuations	Prognostic	[69]
Deshpande et al. *	2016	Prospective correlational	1b	Yes	Preoperative fMRI activation in angular, cingulate gyri and prefrontal cortex	Predictive	[83]

* A thesis in 2014 followed by a paper; both were captured during the screening and data analysis. Both had the same objectives and the same functional biomarker outcomes.

**Table 3 diagnostics-11-00042-t003:** Review articles of biomarkers related to the inner ear.

Author	Year	Level of Evidence	Met Objectives	BioMarker	Key Source	Classification	Category	
Evans & Halliwell	1999	2a	Yes	Reactive oxygen and nitrogen species	Review of intracellular & extracellular antioxidants	Molecular	Pathophysiology/Therapeutic	[74]
Arnaud et al. *	2014	2a	Partially	CII Ab & cell immunity in relapsing polychondritis	Review of immunity in serum	Molecular	Pathogenic model	[26]
				Matrilin-1 in relapsing polychondritis	Review of immunity in serum	Molecular	Pathogenic model	
Okano **	2014	5	Too broad	Immune markers	Review of immunity in serum	Molecular	Pathophysiology/Therapeutic	[75]
Alawieh et al. ***	2015	1a	Yes	Biomarkers of inflammation	Review of immunity in plasma	Molecular	Diagnostic	[34]
Barozzi et al.	2015	2a	No	Inner ear melanocytes	Review of aggregated inner ear melanocytes papers	Molecular	Diagnostic	[79]
Rüttiger et al. ****	2017	5	Yes	Overview of both functional & molecular [e.g., BDNF]	Evoked potentials + Aggregated IHCs & OHCs biomarkers in animal & human studies	Functional and Molecular	Diagnostic/Pathognomonic	[55]
Mulry & Parham	2020	2a	Yes	Inner ear/preclinical models: (Otoconin 90/95, Otogelin, Otoancorin, Cochlin, α-tectorin, β-tectorin.) Vestibular biomarkers: Cochlin & Otolin-	Review of aggregated data of proteins specific to the inner ear, some of which could be detected outside the inner ear	Molecular	Diagnostic	[19]

* CII Ab = Collagen II Antibodies detected in 33% of the sera of patients with Relapsing Polychondritis (RP). Martrilin-1 has also been detected in the serum of RP patients. ** Included increased serum concentrations of antibodies against CMV, herpes zoster, herpes simplex type 1, influenza B, and mumps in patients with idiopathic sudden deafness [15,16]. Cochlear enhancement on magnetic resonance imaging (MRI) as a potential sign of inflammation in the inner ear in sudden deafness [95], reduced concentration of CD4+and CD8+ in ISSNHL, and a response to recombinant human heat shock protein 70, a non-specific heat shock protein, in 19 of 58 (33%) patients with idiopathic SNHL [96]. *** Studied proteomic inner ear structures profile with around 50 proteins. Plasma samples profiling for Meniere’s disease (MD) showed overexpression of factors H and B, fibrinogens α and γ, β actin, and pigment epithelium-derived factor proteins. Concurrently, there was an under-expression of β-2 lipoprotein I, vitamin D binding protein, and Apo-lipoprotein I proteins compared with controls. Due to the low sample size, this needs further studies for confirmation [35]. **** This study was an overview paper with low evidence; it included mostly functional biomarkers such as cochlear microphonics, otoacoustic emissions, and auditory brainstem responses; moreover, it captured some molecular biomarkers, including corticosteroids and brain-derived neurotrophic factor (BDNF).

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
