# Peer review of "Biomarkers for Inner Ear Disorders: Scoping Review on the Role of Biomarkers in Hearing and Balance Disorders"

_diagnostics, 2020, doi:10.3390/diagnostics11010042_

Round 1

Reviewer 1 Report

This manuscript neatly summarizes the available data for prognostic or diagnostic biomarkers, whether functional or molecular, and excluded preclinical studies (out of necessity).  A separate preclinical scoping review would be very informative, and obviously more challenging, but can inform the field further, and may have already been done by Mulry and Parham 2020 (behind paywall) that focused on inner ear only biomarkers (though some were later to be found outside the inner ear).  This manuscript is more expansive and includes autoantibodies, proteomic studies and markers of inflammation. This manuscript demonstrates the difficulty of assigning the use of functional or molecular biomarkers for otologic or audiologic specific diagnosis and prognosis.

The biggest value of this manuscript is the identification of the challenges and difficulties is stating individual biomarkers are inner ear-specific and therefore clinically relevant, at least with current technologies.  The authors are to be commended for undertaking this project, and I look forward to seeing this topic grow as further studies are conducted as molecular- or functional-only investigations to better understand how to use this growing dataset to be more indicative of inner ear-specific disorders.. 

Minor comments

Line 121 ad 129, reviewed studies = 34, yet Figure 1 and the abstract says 43, please clarify

The tables are an especially informative source of human studies conducted to date.

A point to emphasize more strongly is that many of the pro-inflammatory biomarkers are important, yet are very non-specific with respect to etiology, e.g., CRP, and should be used only in context-specific situations, such as SSNHL, and not as inner ear diagnostics when systemic sepsis is present, for example. 

The use of caution by the authors with respect to these and other candidate biomarkers is clinical/ontological use is noteworthy and to be applauded.

B-actin comments are true, yet, b-actin is ubiquitous, and therefore elevated serum levels should only be used as indicative of potential inner ear disorders, along with other more systemic challenges associated with b-actin mutations on a more organismal level, e.g., lymphoma.  This comment may also be true of some other putative biomarkers mentioned in the manuscript, and those can be considered in the light of this comment as well.  In some cases, the authors already did this in this manuscript.

Author Response

This manuscript neatly summarizes the available data for prognostic or diagnostic biomarkers, whether functional or molecular, and excluded preclinical studies (out of necessity).  A separate preclinical scoping review would be very informative, and obviously more challenging, but can inform the field further, and may have already been done by Mulry and Parham 2020 (behind paywall) that focused on inner ear only biomarkers (though some were later to be found outside the inner ear).  This manuscript is more expansive and includes autoantibodies, proteomic studies and markers of inflammation. This manuscript demonstrates the difficulty of assigning the use of functional or molecular biomarkers for otologic or audiologic specific diagnosis and prognosis.

The biggest value of this manuscript is the identification of the challenges and difficulties is stating individual biomarkers are inner ear-specific and therefore clinically relevant, at least with current technologies.  The authors are to be commended for undertaking this project, and I look forward to seeing this topic grow as further studies are conducted as molecular- or functional-only investigations to better understand how to use this growing dataset to be more indicative of inner ear-specific disorders.. 

Many thanks for a great review, which is leveraging the work's end result to a higher academic level. 

Minor comments

Line 121 ad 129, reviewed studies = 34, yet Figure 1 and the abstract says 43, please clarify

The correct placement table has been inserted, and the abstract and results section are reflecting the correct numbers now.

The tables are an especially informative source of human studies conducted to date.

A point to emphasize more strongly is that many of the pro-inflammatory biomarkers are important, yet are very non-specific with respect to etiology, e.g., CRP, and should be used only in context-specific situations, such as SSNHL, and not as inner ear diagnostics when systemic sepsis is present, for example. 

In full agreement with this comment, we emphasized in the discussion the importance of the inner ear specificity in the biomarker discovery.

“Moreover, many currently identified biomarkers are impractical to use in a clinical context, being non-specific (e.g., beta-actin or proinflammatory cytokines),”

“ Furthermore, any non-inner ear-related or non-autoimmune inflammatory condition (e.g., the common cold), as well as the aging process, could influence the inflammatory markers' diagnostic value for inner ear disorders.”

The use of caution by the authors with respect to these and other candidate biomarkers is clinical/ontological use is noteworthy and to be applauded.

B-actin comments are true, yet, b-actin is ubiquitous, and therefore elevated serum levels should only be used as indicative of potential inner ear disorders, along with other more systemic challenges associated with b-actin mutations on a more organismal level, e.g., lymphoma.  This comment may also be true of some other putative biomarkers mentioned in the manuscript, and those can be considered in the light of this comment as well.  In some cases, the authors already did this in this manuscript.

We appreciate this comment and now write in the discussion and conclusions:

“Moreover, many currently identified biomarkers are impractical to use in a clinical context, being non-specific (e.g., beta-actin or proinflammatory cytokines)”

“The majority of biomarkers that can be measured in peripheral blood are inflammatory (e.g., cytokines) or not restricted to the inner ear (e.g. beta-actin) and, therefore, non-specific.”

Reviewer 2 Report

A very interesting manuscript, which brings a new vision to the field that until now has focused mainly on the study of genetic diagnostic factors.

However, it needs some changes in depth in the way of presenting the results and in the discussion. Specifically:

It is important in the field of hearing loss to have diagnostic markers. Obviously, the importance relates to the simplicity of sampling, and therefore it is essential to know the source of the material in which the marker is found.
In this sense, in all Tables the key source column is missing and should be included and discussed.
Likewise, the Tables are, at least in the pdf I have seen, formatted in an unfriendly way for the reader. Information is hard to find and compare and must be reviewed.

The discussion is lengthy at times, and revisits the results, including information that should be included under the Results section and used to present and explain the contents of the Tables. The discussion should focus on the actual utility of the proposed markers, including whether they can be measured in a simple to obtain biological fluid or tissue sample, i.e. nasal epithelium. In this sense, a selection of markers with real potential to be used for diagnosis should be shown. Finally, this selection should be connected to other evidence, including complementary genetic and experimental models. The specific conclusions of the study are not clear.

Author Response

A very interesting manuscript, which brings a new vision to the field that until now has focused mainly on the study of genetic diagnostic factors.

However, it needs some changes in depth in the way of presenting the results and in the discussion. Specifically:

It is important in the field of hearing loss to have diagnostic markers. Obviously, the importance relates to the simplicity of sampling, and therefore it is essential to know the source of the material in which the marker is found.
In this sense, in all Tables the key source column is missing and should be included and discussed.
Likewise, the Tables are, at least in the pdf I have seen, formatted in an unfriendly way for the reader. Information is hard to find and compare and must be reviewed.

Thank you for a very valid point. An additional column titled the key source tables 1& 3. The orientation of all tables has been changed to landscape orientation to improve readability.

The discussion is lengthy at times, and revisits the results, including information that should be included under the Results section and used to present and explain the contents of the Tables.

Point taken with thanks. The details of biomarkers have been moved to the results section. This way, the discussion has become more of a collective and concise information

The discussion should focus on the actual utility of the proposed markers, including whether they can be measured in a simple to obtain biological fluid or tissue sample, i.e. nasal epithelium. In this sense, a selection of markers with real potential to be used for diagnosis should be shown. Finally, this selection should be connected to other evidence, including complementary genetic and experimental models.

The discussion was revised and focuses now on the utility of the proposed markers and the feasibility of obtaining the inner ear samples.

The specific conclusions of the study are not clear.

Future directions and conclusions have been revisited.